# Remote immune processes revealed by immune-derived circulating cell-free DNA

Ilana Fox-Fisher[1], Sheina Piyanzin[1], Bracha Lea Ochana[1], Agnes Klochendler[1], Judith Magenheim[1], Ayelet Peretz[1], Netanel Loyfer[2], Joshua Moss[1], Daniel Cohen[1], Yaron Drori[3], Nehemya Friedman[3], Michal Mandelboim[3], Marc E Rothenberg[4], Julie M Caldwell[4], Mark Rochman[4], Arash Jamshidi[5], Gordon Cann[5], David Lavi[6], Tommy Kaplan[2,7], Benjamin Glaser[8], Ruth Shemer[1], Yuval Dor[1]*

[1]Department of Developmental Biology and Cancer Research, The Institute for Medical Research, Israel-Canada, The Hebrew University-Hadassah Medical School, Jerusalem, Israel; [2]School of Computer Science and Engineering, The Hebrew University of Jerusalem, Jerusalem, Israel; [3]Department of Epidemiology and Preventive Medicine, School of Public Health, Sackler Faculty of Medicine, Tel-Aviv University, Tel-Aviv, Israel, and Central Virology Laboratory, Ministry of Health, Chaim Sheba Medical Center, Ramat-Gan, Israel; [4]Division of Allergy and Immunology, Department of Pediatrics, Cincinnati Children's Hospital Medical Center, University of Cincinnati, Cincinnati, United States; [5]GRAIL, Menlo Park, United States; [6]Department of Hematology, Hadassah Hebrew University Medical Center, Jerusalem, Israel; [7]Department of Developmental Biology and Cancer Research, The Institute for Medical Research, The Hebrew University-Hadassah Medical School, Jerusalem, Israel; [8]Endocrinology and Metabolism Service, Hadassah Hebrew University Medical Center, Jerusalem, Israel

**Abstract** Blood cell counts often fail to report on immune processes occurring in remote tissues. Here, we use immune cell type-specific methylation patterns in circulating cell-free DNA (cfDNA) for studying human immune cell dynamics. We characterized cfDNA released from specific immune cell types in healthy individuals (N = 242), cross sectionally and longitudinally. Immune cfDNA levels had no individual steady state as opposed to blood cell counts, suggesting that cfDNA concentration reflects adjustment of cell survival to maintain homeostatic cell numbers. We also observed selective elevation of immune-derived cfDNA upon perturbations of immune homeostasis. Following influenza vaccination (N = 92), B-cell-derived cfDNA levels increased prior to elevated B-cell counts and predicted efficacy of antibody production. Patients with eosinophilic esophagitis (N = 21) and B-cell lymphoma (N = 27) showed selective elevation of eosinophil and B-cell cfDNA, respectively, which were undetectable by cell counts in blood. Immune-derived cfDNA provides a novel biomarker for monitoring immune responses to physiological and pathological processes that are not accessible using conventional methods.

*For correspondence:
yuvald@ekmd.huji.ac.il

## Introduction

Circulating biomarkers for monitoring inflammatory or immune responses are an essential part of diagnostic medicine and an important tool for studying physiological and pathological processes. These include, among others, counts of specific immune cell types in peripheral blood, RNA expression profiles in blood cells (*Maas et al., 2002*; *Tuller et al., 2013*), and levels of circulating proteins

such as C-reactive protein (CRP) (*Gabay and Kushner, 1999*; *Sproston and Ashworth, 2018*). A major limitation of circulating immune cell analysis is that it often fails to report on immune processes taking place in remote locations. Conversely, CRP and similar proteins do reflect the presence of tissue inflammation but are highly non-specific with regard to tissue location and the nature of inflammatory process (*Gabay and Kushner, 1999*).

Dying cells release nucleosome-size fragments of cell-free DNA (cfDNA), which travel transiently in blood before being cleared by the liver (*Heitzer et al., 2019*). Analysis of the sequence of such fragments is emerging as a powerful diagnostic modality. Liquid biopsies using cfDNA have been applied to reveal the presence of mutations in a fetus as reflected in maternal cfDNA (*Bianchi et al., 2014*; *Fan et al., 2012*; *Lo et al., 1997*), identify and monitor tumor dynamics via the presence of somatic mutations in plasma (*Wan et al., 2017*), and detect the rejection of transplanted organs when the levels of donor-derived DNA markers are elevated in recipient plasma (*De Vlaminck et al., 2014*; *De Vlaminck et al., 2015*). More recently, we and others have shown that tissue-specific DNA methylation patterns can be used to determine the tissue origins of cfDNA, allowing to infer cell death dynamics in health and disease even when no genetic differences exist between the host and the tissue of interest (*Cheng et al., 2019*; *Lehmann-Werman et al., 2016*).

Although the majority of cfDNA in healthy individuals is known to originate in hematopoietic cells (*Lehmann-Werman et al., 2016*; *Moss et al., 2018*; *Sun et al., 2015*), it has often been regarded as background noise, against which one may look for rare cfDNA molecules released from a solid tissue of interest. We hypothesized that identification of immune cell-derived cfDNA could open a window into immune and inflammatory cell dynamics, even in cases where peripheral blood counts are not informative. Here, we describe the development of a panel of immune cell type-specific DNA methylation markers, and the use of this panel for cfDNA-based assessment of human immune cell turnover in health and disease. We show that immune cell cfDNA measurement can provide clinical biomarkers in multiple disease and treatment conditions, otherwise undetectable by cell subset enumeration in blood.

## Results

### Identification of cell type-specific DNA methylation markers for immune cells

Using a reference methylome atlas of 32 primary human tissues and sorted cell types (*Moss et al., 2018*), we searched for CpG sites that are uniquely methylated or unmethylated in a specific immune cell type. Notably, across the entire atlas, the vast majority of such unique loci are hypomethylated in the cell type of interest and methylated elsewhere (typically marking cell type-specific enhancers), while just a small minority are methylated in a given cell type and hypomethylated elsewhere (manuscript in preparation). We identified dozens of uniquely hypomethylated CpG sites for most cell types examined, qualifying these as biomarkers for DNA derived from a given cell type (*Figure 1A*). Based on this in silico comparative analysis, we selected for further work 17 different CpG sites, whose combined methylation status could distinguish the DNA of seven major immune cell types: neutrophils, eosinophils, monocytes, B-cells, CD3 T-cells, CD8 cytotoxic T-cells, and regulatory T-cells (Tregs). For each marker CpG we designed PCR primers to amplify a fragment of up to 160 bp flanking it, considering the typical nucleosome size of cfDNA molecules. Amplicons included additional adjacent CpG sites, to gain enhanced cell type specificity due to the regional nature of tissue-specific DNA methylation (*Lehmann-Werman et al., 2016*). We then established a multiplex PCR protocol, to co-amplify all 17 markers from bisulfite-treated DNA followed by next-generation sequencing (NGS) for assessment of methylation patterns (*Neiman et al., 2020*). Methylation patterns of amplified loci across 18 different human tissues validated the patterns inferred from in silico analysis and supported the ability of this marker cocktail to specifically identify the presence of DNA from each of the seven immune cell types (*Figure 1B*). We also assessed assay sensitivity and accuracy via spike-in experiments. We mixed human leukocyte DNA with DNA from the HEK-293 human embryonic kidney cell line and used the methylation cocktail to assess the fraction of each immune cell type. The markers quantitatively detected the presence of DNA from specific immune cell types even when blood DNA was diluted 10- to 20-fold (*Figure 1C* and *Figure 1—figure supplement 1*). These findings establish

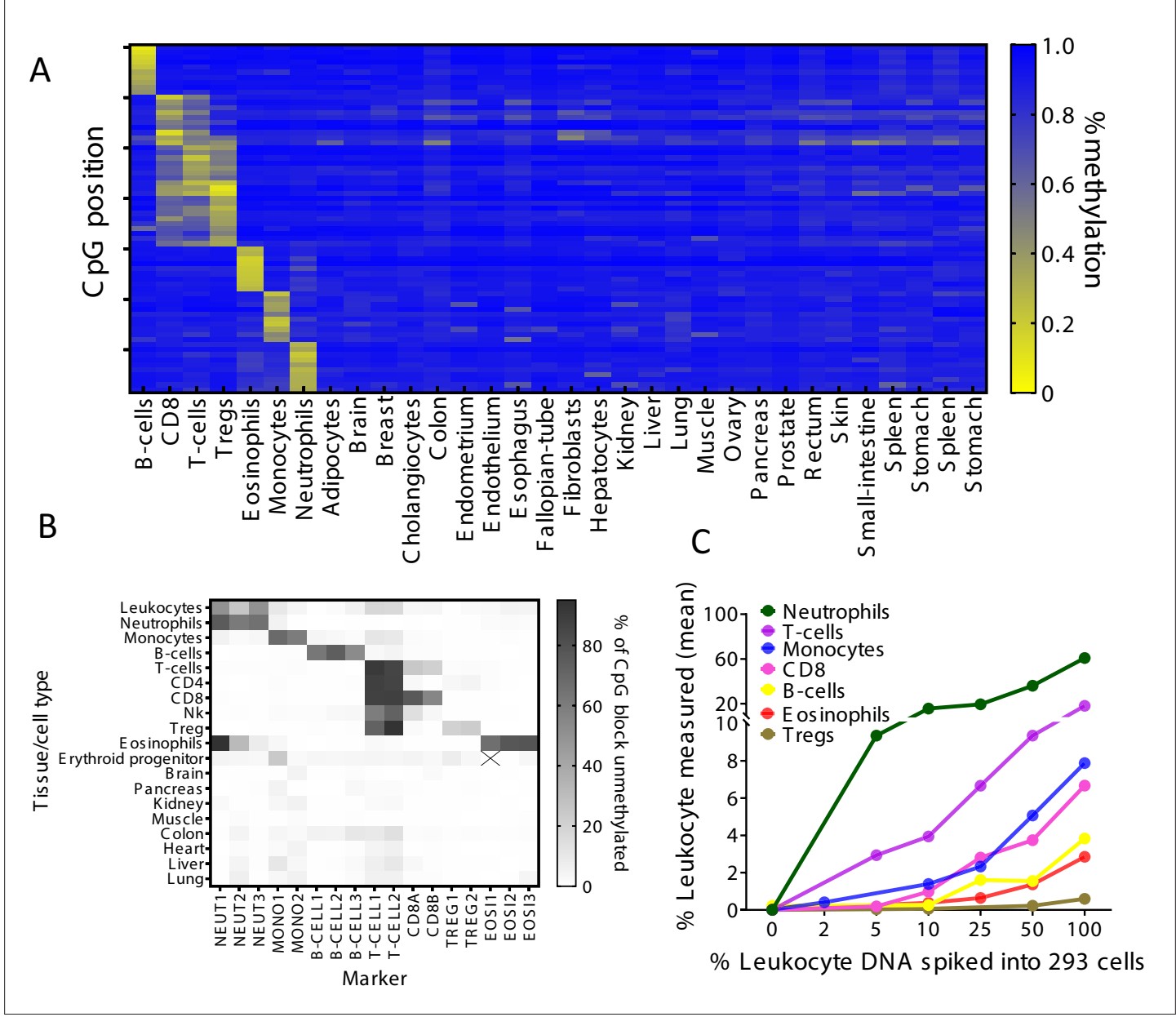

**Figure 1.** Identification of specific immune cell DNA methylation markers. (**A**) Methylation atlas, based on Illumina 450K arrays, composed of 32 tissues and sorted cells (columns). For each immune cell type we chose the top 10 CpGs that are hypomethylated (yellow) in the specific immune cell type and hypermethylated (blue) in other tissues and cells. This yielded 70 cell-specific CpG sites (rows) for seven different immune cell subtypes – B-cells, CD8 cytotoxic T-cells, CD3 T-cells, regulatory T-cells, eosinophils, monocytes, and neutrophils. (**B**) Methylation patterns of 17 loci, selected from the 70 shown in panel A, based on the presence of multiple adjacent hypomethylated CpGs within an amplicon of up to 160 bp. Each methylation marker (columns) was assessed using genomic DNA from 19 different tissues and cell types (rows). All 17 markers were amplified in one multiplex PCR. Shades of gray represent the percentage of fully unmethylated molecules from the indicated marker in DNA from the indicated cell type. (**C**) Spike-in experiments assessing assay sensitivity. Human leukocyte DNA was mixed with DNA from HEK-293 cells (human embryonic kidney cells) in the indicated proportions. Colored lines show the inferred percentage of DNA from the indicated immune cell type in the mixture, as a function of the percentage of leukocyte DNA in the mixture. The percentage of DNA from each immune cell type was calculated using markers specific to neutrophils (NEUT1, NEUT2, NEUT3), monocytes (MONO1, MONO2), eosinophils (EOSI2, EOSI3), B-cells (B-CELL1, B-CELL2, B-CELL3), CD3 T-cells (T-CELL1, T-CELL2), CD8 cytotoxic T-cells (CD8A, CD8B), and regulatory T-cells (TREG1, TREG2).

The online version of this article includes the following figure supplement(s) for figure 1:

**Figure supplement 1.** Identification of specific immune cell DNA methylation markers.

specificity, sensitivity, and accuracy of the methylation marker cocktail for detection of DNA derived from the seven selected immune cell types.

## Immune cfDNA reflects cell turnover rather than counts of circulating blood cells

To validate assay accuracy, we applied the immune cell methylation markers to genomic DNA of blood cells, expecting to observe signals that agree with cell ratios as determined by complete blood counts (CBC). We obtained 392 blood samples from 79 healthy individuals at different time points, and simultaneously tested for CBC, and methylation marker cocktail both in DNA extracted from whole blood and in cfDNA extracted from plasma. Theoretically, immune cell cfDNA could be a mere reflection of the counts of each cell type (e.g. if it is released mostly from blood cells that have died during blood draw or preparation of plasma). In such a case, immune cfDNA should correlate well with CBC (and with immune methylation markers measured in genomic DNA from whole blood). Alternatively, if cfDNA reflects cell death events that took place in vivo, the correlation to cell counts is expected to be weaker.

Comparing the CBC to DNA methylation pattern, we observed a strong correlation between assessments of specific cell fractions in the two methods (Pearson's correlations; r = 0.67–0.83, p-value < 0.0001, *Figure 2A* and *Figure 2—figure supplement 1*), supporting validity of the methylation assay for identifying fractions of DNA derived from specific immune cell types, consistent with previous findings (*Baron et al., 2018*). However, comparing cfDNA methylation markers in plasma to blood DNA methylation markers and CBC, we observed that proportion of cfDNA from specific immune cell types did not correlate with the proportion of the same markers in circulating blood cells and with CBC (Pearson's correlations; r = 0.14–0.53, *Figure 2B–C* and *Figure 2—figure supplement 1*). These findings suggest that immune cfDNA levels are the result of biological processes beyond immune cell counts.

We reasoned that over- or under-representation of DNA fragments from a specific immune cell type in plasma compared with blood counts most likely result from differences in cell turnover. The concentration of cfDNA from a given cell type should be a function of the total number of cells that have died per unit time (turnover rate), which is derived from the number of cells and their lifespan:

$$\frac{\text{Total cell number}}{\text{Lifespan}} = \text{Turnover rate}$$

The larger the population of a given cell type (both circulating and tissue-resident), the more cfDNA it will release; similarly, the shorter is lifespan, the more DNA will be released to plasma. The cfDNA findings were consistent with this model. For example, the fraction of lymphocyte cfDNA in plasma was always smaller than the fraction of lymphocyte DNA in circulating blood cells or the fraction of lymphocytes in CBC (*Figure 2B–C* and *Figure 2—figure supplement 1*), in agreement with the long half-life of lymphocytes compared with other blood cell types (*Macallan et al., 2005*; *Michie et al., 1992*). Conversely, the fraction of monocyte cfDNA was larger than the fraction of monocyte DNA in genomic DNA from whole blood or the fraction of monocytes in CBC (*Figure 2B–C*), consistent with the shorter half-life of monocytes (*Patel et al., 2017*).

To further examine the relative presence of immune cfDNA in plasma and whole blood, we employed an independent set of samples and an independent technology to measure and interpret methylation markers. Specifically, we performed deconvolution (*Moss et al., 2018*) of methylomes obtained by whole genome bisulfite sequencing (WGBS) (85× average coverage), on genomic DNA from whole blood, and matched plasma cfDNA from 23 healthy donors (see Materials and methods). This analysis, based on genome-wide methylation patterns, also revealed that lymphocyte and monocyte cfDNA was under- and over-represented, respectively, relative to the abundance of DNA from these cells in blood (lymphocytes, p-value = 0.002; monocytes, p-value = 0.0005, Kruskal-Wallis) (*Figure 2D*).

These findings support the exciting idea that cfDNA levels from a given immune cell type integrate total cell number and the lifespan of that cell type, and can provide information on processes not evident from circulating cell counts. For example, if the level of cfDNA from a specific immune cell type increases while the circulating counts of this cell type are unchanged, this can indicate either growth in the size of a tissue-resident population, or increased cell turnover, both of which are important

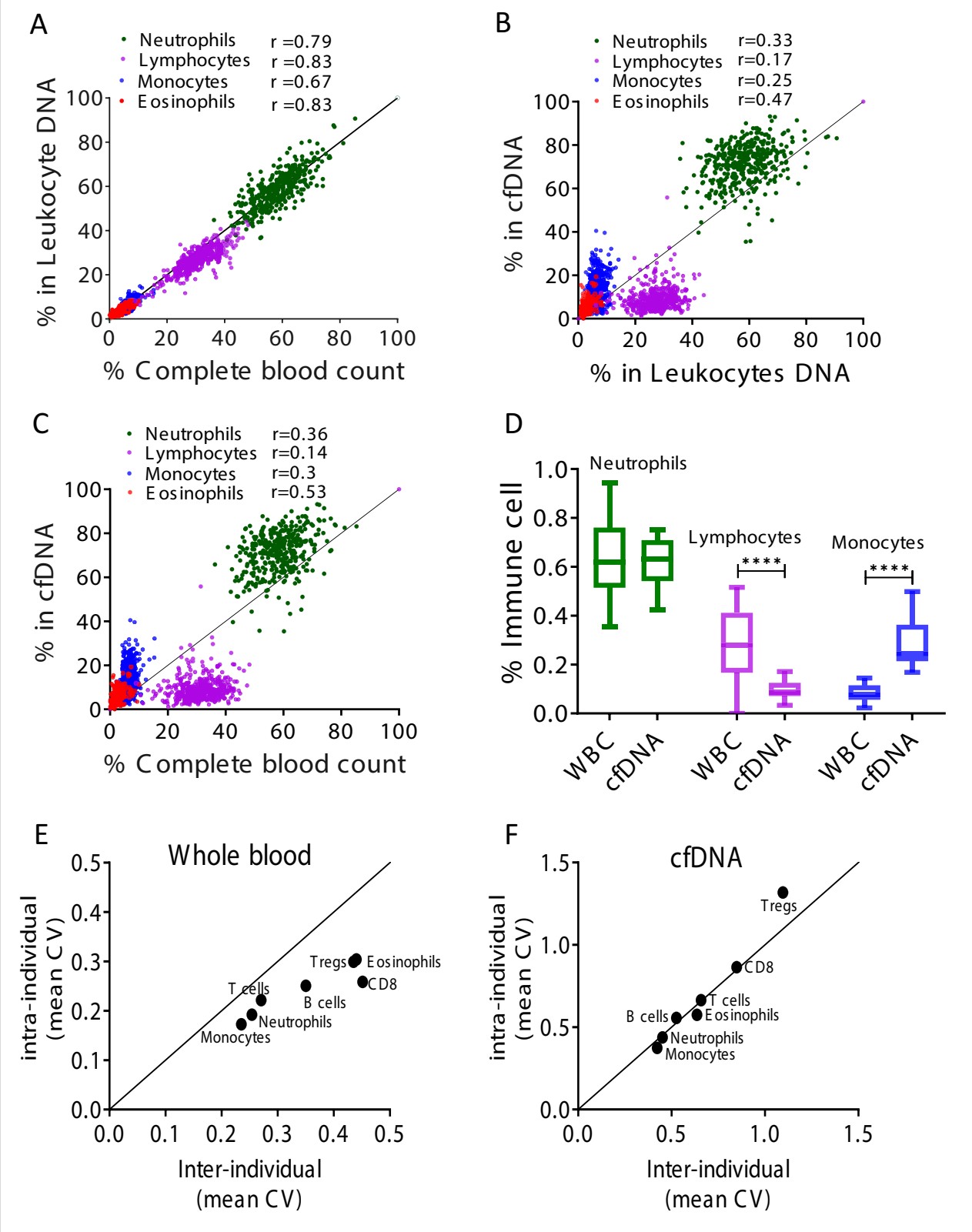

**Figure 2.** Immune cell-free DNA (cfDNA) methylation markers distribute differently than circulating immune cells and reflect immune cell turnover. (**A**) Levels of immune methylation markers in genomic DNA extracted from whole blood, versus complete blood counts (CBC) from same donors. A total of 392 plasma and blood samples were obtained from 79 healthy donors (47 females, 32 males; age range 20–73). Shown are Pearson's correlations; p-value < 0.0001. (**B**) cfDNA methylation versus whole blood methylation of the same donors. (**C**) cfDNA methylation versus CBC of the same donors.

*Figure 2 continued on next page*

*Figure 2 continued*

Note that cfDNA proportions of immune cells differ from the proportions of these cell types in peripheral blood. (**D**) Deconvolution of cfDNA and white blood cell (WBC) methylomes generated using whole genome bisulfite sequencing (WGBS) of 23 healthy donors. Note under-representation of lymphocyte DNA and over-representation of monocyte DNA in cfDNA compared with blood DNA (lymphocytes, p-value = 0.0021; monocytes, p-value = 0.0005, Kruskal-Wallis). (**E–F**) XY scatter plots showing the average of inter-individual coefficient of variation (X-axis) and intra-individual coefficient of variation (Y-axis) for each immune cell type in whole blood (**E**) and in cfDNA (**F**) based on methylation markers. Black line represents perfect correlation between inter- and intra-individual dual variation. Dots below the black line reflect greater inter-individual variation and dots that are above reflect greater intra-individual variance. A smaller intra-individual variation in whole blood suggests a set point for proportions of blood cell types in each individual. By contrast, cfDNA levels of immune markers vary similarly within and between individuals.

The online version of this article includes the following figure supplement(s) for figure 2:

**Figure supplement 1.** Levels of immune cell-derived cell-free DNA (cfDNA) are not correlated with the counts of these cells in circulation, and reflect cell type-specific turnover rates.

**Figure supplement 2.** Immune-derived cell-free DNA (cfDNA) characterization in a healthy population.

immune parameters that cannot be easily obtained otherwise. Below, we provide evidence that such information can be extracted following perturbations of immune homeostasis.

## Estimation of immune cfDNA baseline in healthy individuals

To define the baseline levels of immune-derived cfDNA in healthy individuals, we collected and tested plasma samples from 227 healthy donors (males and females, ages 1–85 years). Consistent with our previous plasma methylome analysis (*Moss et al., 2018*), we found that neutrophils were the main source of immune-derived cfDNA (mean = 390, range 10–1064 genome equivalents [GE]/ml), followed by monocytes (mean = 101, range 6–233 GE/ml), eosinophils (mean = 38, range 0–111 GE/ml) and lymphocytes (T-cells, mean = 30, range 1-79; B-cells, mean = 17, range 3–42; CD8 T-cells, mean = 8, range 0–27; Tregs, mean = 2, range 0–9 GE/ml) (*Figure 2—figure supplement 2*). These data chart the normal range of cfDNA concentrations from specific immune cell types, including age and gender characteristics, against which pathological deviations can be identified (see below).

We also conducted a longitudinal study, to understand how immune-derived cfDNA is changing over time in the same individual. We collected weekly blood samples from 15 healthy donors over a period of 6 weeks. For each sample we obtained CBC and measured immune DNA methylation markers in genomic DNA of blood cells and in plasma cfDNA. We then calculated the coefficient of variation among CBC, blood methylation markers, and cfDNA methylation markers, within and between individuals. In circulating blood cells, the inter-individual variation in immune methylation markers and CBC was always higher than the intra-individual variation in these markers (*Figure 2E* and *Figure 2—figure supplement 2*). This is consistent with previous reports that blood cell counts among individuals are more similar to themselves than to others, indicating distinct set points per person for the total number of specific immune cell types circulating in blood (*Alpert et al., 2019*; *Carr et al., 2016*). Strikingly, cfDNA values of the same immune methylation markers varied to the same extent among samples of the same individual and among samples of different individuals (*Figure 2F*). This argues that unlike cell counts, cfDNA of immune cells has no individual set point. Rather, cfDNA levels appear to reflect homeostatic maintenance of cell number, whereby cell birth and death are modulated to maintain a desired cell count.

## Elevation of B-cell-derived cfDNA after influenza vaccination precedes changes in cell counts and correlates with specific antibody production

We hypothesized that upon perturbations of the immune system, cfDNA markers will reveal information about immune cell dynamics that is not present in peripheral blood cell counts, for example, extensive cell death during the process of affinity maturation, which repeatedly selects for B-cell clones with increased antibody-target affinity. To test this hypothesis, we examined longitudinal blood samples from healthy individuals who received an annual quadrivalent influenza vaccination (*Nakaya et al., 2011*; *Voigt et al., 2018*). The influenza vaccine response is mediated mostly by the humoral immune system (B-cells) aided by CD4 T-cells (*Gage et al., 2018*). Changes in circulating cell counts occur a week after vaccination, reflecting processes such as plasma cell formation (*Victora and Wilson, 2015*). cfDNA responses to vaccination were not previously reported. We recruited 92 healthy volunteers (age range 20–73, mean age 37.4) who received the vaccination in 2018 or 2019.

From each volunteer we obtained blood samples a day before vaccination (day 0, D0), and at day 3, 7, and 28 post-vaccination. Consistent with previous reports, B-cell counts (measured by methylation analysis of DNA from whole blood) were moderately but significantly elevated on day 7, and persisted to day 28 (p-value = 0.0048, Kruskal-Wallis) (*Figure 3A*; *Li et al., 2012*). Surprisingly, B-cell-derived cfDNA levels increased as early as day 3, peaked on day 7 and returned to baseline levels on day 28 (p-value < 0.0001, Kruskal-Wallis) (*Figure 3B and D*), suggesting that cfDNA reveals an early increase in the turnover of B-cells following vaccination, which is not portrayed in circulating B-cells. We observed a similar trend in the ratio of B-cell cfDNA to B-cell counts in each individual (*Figure 3C* and p-value=0.016, Kruskal-Wallis, B-cell counts calculated from methylation markers in whole blood). Of note, this response was specific to B-cell-derived cfDNA; total cfDNA levels did not change over the time course of vaccination, nor did cfDNA levels of other immune cell types (*Figure 3—figure supplement 1*). Taken together, this strengthens evidence that cfDNA changes reflect processes beyond alterations in absolute circulating cell counts in a cell-specific manner.

To ask if the elevation of B-cell cfDNA has functional significance in the development of an immune response, we obtained information on the production of antibodies. We classified all volunteers into responders or non-responders according to the hemagglutinin antibody titer measured at 28 days post-vaccination, and asked if B-cell cfDNA or B-cell counts correlated with antibody production. Responders had a significantly higher peak elevation of B-cell cfDNA relative to their pre-vaccination baseline levels compared with non-responders (p-value = 0.044, Mann-Whitney, AUC = 0.7, p-value = 0.04) (*Figure 3E and F* and *Figure 3—figure supplement 2*). Peripheral B-cell counts were not different between responders and non-responders (p-value = 0.2, Mann-Whitney) (*Figure 3G*). It is well established that influenza vaccination is more effective in younger individuals (*Del Giudice et al., 2015*; *Ranjeva et al., 2019*; *Siegrist and Aspinall, 2009*; *Wagner et al., 2018*). To examine the relationship between age, antibody production and cfDNA we plotted the fold elevation of B-cell cfDNA from baseline as a function of donor age, and marked responders and non-responders. Non-responders to vaccination in our cohort were all above 35 years and tended to have a minimal elevation of B-cell cfDNA above baseline even when compared to people in their age group (*Figure 3H* and *Figure 3—figure supplement 2*, peak elevation of B-cell cfDNA in responders versus non-responders p-value = 0.089), suggesting that B-cell cfDNA dynamics report on a biological process independent of age. We conclude that B-cell turnover (as reflected in B-cell cfDNA but not in B-cell counts) captures an early response of the immune system to influenza vaccination that predicts a functional outcome, suggesting cell-specific cfDNA could serve as a sensitive biomarker of functional immune changes.

## Selective elevation of eosinophil-derived cfDNA in patients with eosinophilic esophagitis

To test the hypothesis that immune-derived cfDNA can reveal pathological inflammatory processes in remote locations, we studied patients with eosinophilic esophagitis (EoE). EoE is a chronic inflammatory disease characterized clinically by esophageal dysfunction, and histologically by eosinophil-predominant inflammation of the esophagus (*Liacouras et al., 2011*). Diagnosis of EoE requires an invasive endoscopic biopsy. Notably, most patients do not show peripheral eosinophilia (*Aceves et al., 2007*; *Dellon et al., 2009*). We analyzed blindly immune cfDNA markers in plasma samples from patients with active EoE (N = 21), patients with EoE in remission (N = 24), and healthy controls (N = 14). Patients with active EoE had elevated levels of eosinophil cfDNA (mean = 115 GE/ml) compared with both healthy controls (mean = 34 GE/ml, p-value = 0.0056) and patients with inactive EoE (mean = 36 GE/ml, p-value = 0.0003, Kruskal-Wallis), while other immune cfDNA markers were not elevated in active EoE patients (*Figure 4A and B* and *Figure 4—figure supplement 1*). The fraction of eosinophils in blood was not significantly elevated in EoE patients (*Figure 4C* and p-value=0.1, Kruskal-Wallis), consistent with restriction of eosinophil abundance to the esophagus and further supporting the idea that immune cfDNA is not a reflection of circulating immune cells.

Among a small subset of donors for which we had access to plasma, PBMC and CBC (12 active EoE, 8 inactive EoE, 3 controls), elevated eosinophil counts and elevated eosinophil cfDNA levels were observed in non-overlapping groups of EoE patients (elevated eosinophil counts in 4/12 patients with active EoE, 2/8 patients with inactive EoE, as previously reported [*Dellon et al., 2009*]; elevated eosinophil cfDNA in 5/12 patients with active EoE), suggesting that counts and cfDNA reflect different biological processes (*Figure 4D*). Finally, we generated receiver operating characteristic (ROC) curves

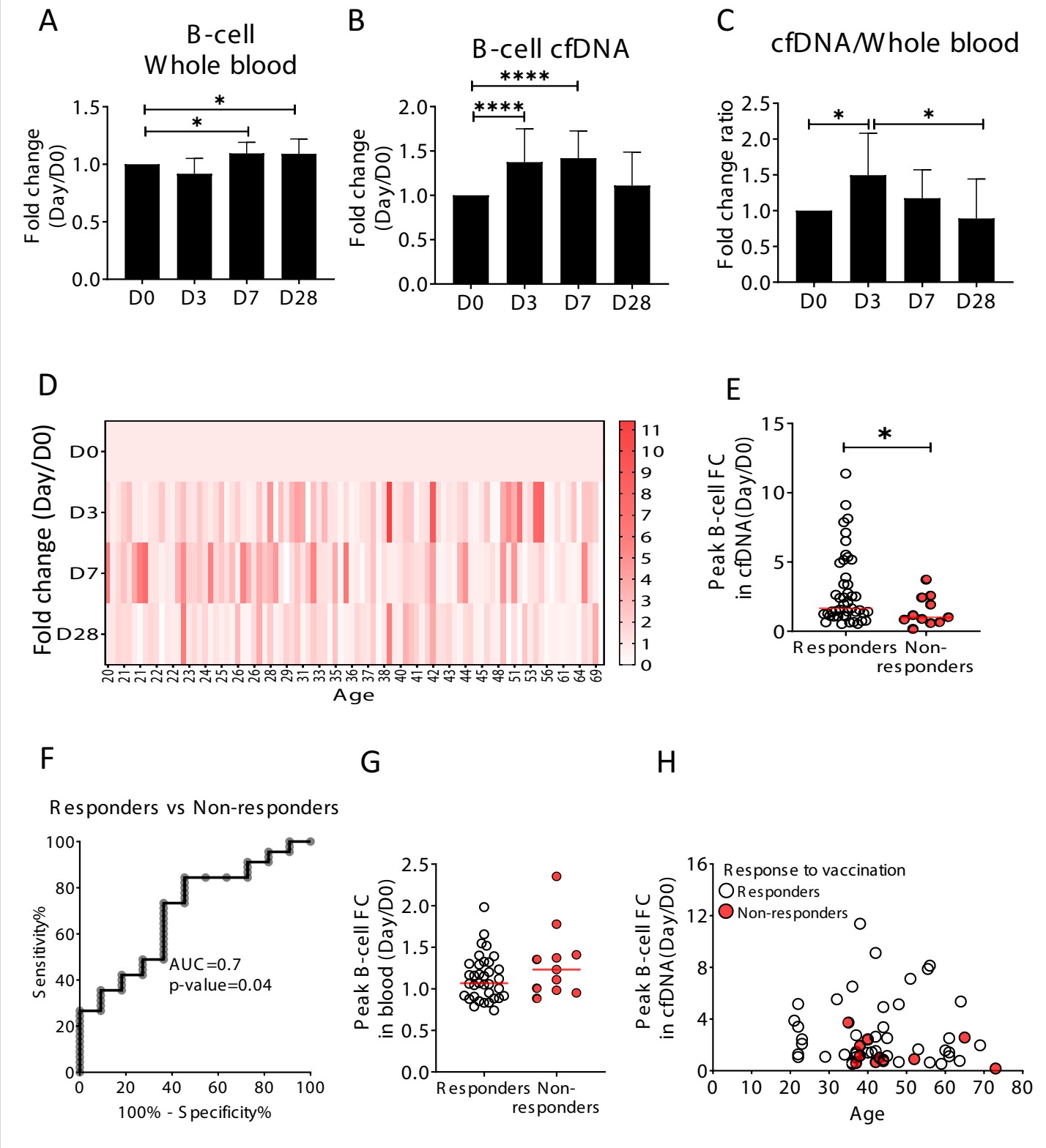

**Figure 3.** Elevation of B-cell-derived cell-free DNA (cfDNA) after influenza vaccination precedes changes in cell counts and reflects efficacy of response to vaccination. Plasma, serum, and blood samples were obtained from 92 healthy donors receiving the influenza vaccination (55 females, 37 males, age range 20–73 years). (**A**) Circulating B-cells, assessed using methylation markers, are elevated on days 7 (p-value = 0.03) and 28 (p-value = 0.021) compared to baseline (Kruskal-Wallis) (N = 64). (**B**) B-cell-derived cfDNA markers were normalized to the levels of each individual at baseline (D0) and represented as fold change. B-cell-derived cfDNA is elevated compared to baseline on days 3 and 7 after influenza vaccination (p-value < 0.0001,

*Figure 3 continued on next page*

*Figure 3 continued*

Kruskal-Wallis) (N = 92). (**C**) The ratio of B-cell-derived cfDNA and B-cells in blood was calculated for each individual. On day 3 the ratio was significantly higher than at baseline (p-value = 0.016). Bars indicate the median and error bars indicate the confidence interval. (**D**) A heat map showing the change of B-cell-derived cfDNA in each individual following vaccination relative to baseline. Donors are ordered by their age. (**E**) Serum antibody titers were used to divide donors into responders (n = 45) who have developed antibodies following vaccination, and non-responders (n = 11) who have not. Graph shows the maximal elevation (fold change [FC] from baseline) in B-cell-derived cfDNA that was recorded for each donor. (p-value = 0.045, Mann-Whitney). (**F**) Receiver operating characteristic (ROC) curve for distinguishing responders and non-responders to vaccination. (**G**) Maximal elevation in B-cell counts in blood based on methylation markers does not differ between responders and non-responders. p-Value = 0.2, Mann-Whitney test. (**H**) XY scatter plot for peak B-cell cfDNA elevation as a function of age. Non-responders (colored with red) tend to be older and show reduced elevation.

The online version of this article includes the following figure supplement(s) for figure 3:

**Figure supplement 1.** Specific elevation of B-cell-derived cell-free DNA (cfDNA) after influenza vaccination, prior to changes in cell counts and in correlation to efficacy of response to vaccination.

**Figure supplement 2.** Specific elevation of B-cell-derived cell-free DNA (cfDNA) after influenza vaccination, prior to changes in cell counts and in correlation to efficacy of response to vaccination.

to test our ability to identify active EoE patients. Eosinophil cfDNA could distinguish active EoE from healthy controls (*Figure 4E*, AUC 0.83, p-value = 0.001) and from patients with inactive disease (*Figure 4F*, AUC 0.84, p-value = 0.0001), with high specificity and sensitivity. These findings suggest that cell type-specific cfDNA can be used to detect clinical inflammation occurring in solid tissues that is not reflected in peripheral cell counts.

## B-cell-derived cfDNA elevation in patients with B-cell lymphoma

Hematological malignancies occurring in remote immune organs such as the bone marrow, spleen, and lymph nodes are often undetectable in peripheral blood (*Conlan et al., 1991*). We reasoned that increased turnover of cancer cells in hematological malignancies would release cfDNA molecules carrying methylation marks of the normal cell type from which the tumor originated, informing on tumor presence and dynamics. In support of this idea, previous plasma methylome analysis by *Sun et al., 2015*, demonstrated elevated B-cell-derived cfDNA in a pregnant woman that unknowingly had B-cell lymphoma. In addition to tumor-derived cfDNA, cell type-specific cfDNA markers could reveal collateral damage incurred by the tumor to normal adjacent cells (*Ménétrier-Caux et al., 2019*; *Ray-Coquard et al., 2009*). To test this idea we examined blood samples from patients with B-cell lymphoma, a disease which often requires imaging and invasive biopsies for diagnosis and monitoring (*Barrington et al., 2014*; *Laurent et al., 2017*). We studied plasma and blood cells from 17 newly diagnosed (treatment-naïve) B-cell lymphoma patients (diffuse large B-cell lymphoma, n = 6; Hodgkin's lymphoma, n = 5; follicular lymphoma, n = 6) and age-matched healthy controls (*Supplementary file 1*). Lymphoma patients (mean = 264.4 GE/ml) had dramatically elevated levels of B-cell-derived cfDNA compared with controls (mean = 18.3 GE/ml, p-value < 0.0001), while B-cell counts in peripheral blood were actually decreased (control; mean = 0.162, lymphoma; mean = 0.079, $10^9$/l, p-value = 0.0059, Mann-Whitney) (*Figure 5A–C*). We observed that the level of B-cell cfDNA accurately distinguished B-cell lymphoma patients from healthy controls, much better than did B-cell counts (cfDNA, AUC = 0.98, p-value < 0.0001; B-cell counts, AUC = 0.75, p-value = 0.006; *Figure 5D and E*). Total levels of cfDNA as well as the levels of other immune cfDNA markers were also elevated in lymphoma patients, consistent with reports on alterations in non-B-cells in lymphoma (*Simone, 2013*). We observed the strongest response in the levels of B-cell cfDNA (14.4-fold increase compared with controls), CD8 cytotoxic T-cells (10.7-fold), and Tregs (13.8-fold) (*Figure 5F*, *Figure 5—figure supplement 1*). Lymphocyte counts were decreased, such that the ratio of cfDNA to cell count for each cell type was dramatically elevated in lymphoma patients (*Figure 5G*). In an additional cohort of lymphoma patients that were monitored before and after treatment, we observed a decrease in B-cell-derived cfDNA in most patients (n = 16, p-value = 0.0032, Kruskal-Wallis test), similarly to what have been observed in other cancers following treatment (*Moss et al., 2020*; *Figure 5H*). Initial analysis did not reveal a correlation between B-cell cfDNA and clinical outcome of treatment as defined by PET-CT (e.g. the few patients whose B-cell cfDNA were not reduced after treatment did not stand out as having worse prognosis), suggesting more complex relationships between B-cell cfDNA dynamics and clinical phenotype. To validate these findings, we performed the analysis on plasma samples from a second, independent cohort of untreated lymphoma patients and healthy controls. As in the first

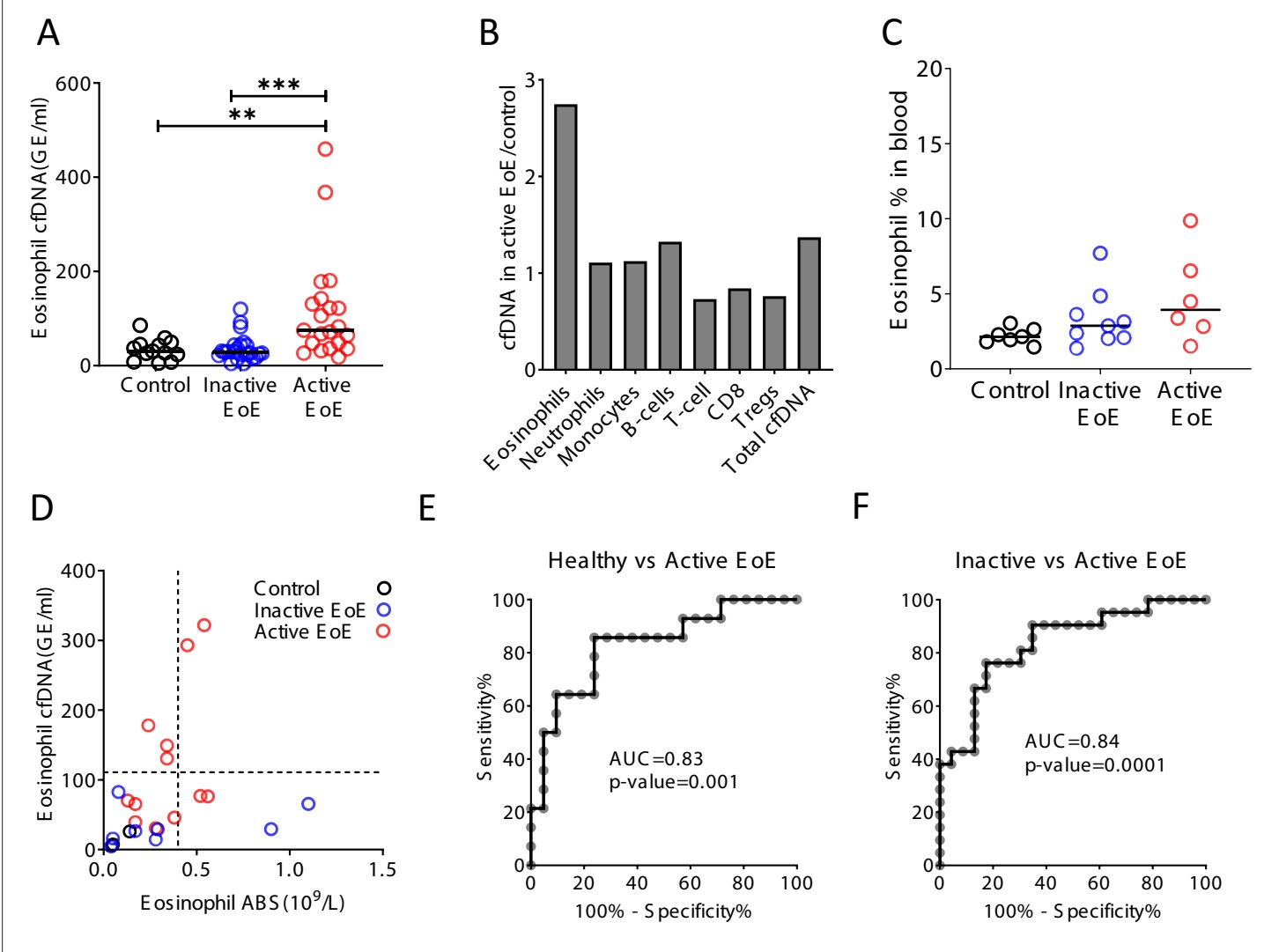

**Figure 4.** Selective elevation of eosinophil-derived cell-free DNA (cfDNA) in patients with eosinophilic esophagitis (EoE). (**A**) Eosinophil-derived cfDNA in active EoE patients (n = 21) compared with healthy controls (n = 14, p-value = 0.0056) and patients with EoE in remission (inactive EoE, n = 24, p-value = 0.0003, Kruskal-Wallis). (**B**) Differences between immune cfDNA populations in active EoE patients and healthy controls (mean active EoE/ mean control). (**C**) Eosinophil DNA markers in whole blood of patients and controls (p-value = 0.1, Kruskal-Wallis test). (**D**) XY scatter plot for eosinophil-derived cfDNA levels versus eosinophil absolute count (ABS) in blood. Dashed lines indicate healthy maximal baseline levels of eosinophil absolute counts in blood, and eosinophil-derived cfDNA in plasma. (**E**) Receiver operating characteristic (ROC) curve for the diagnosis of active EoE, using eosinophil cfDNA levels in plasma of healthy controls and patients with active EoE. (**F**) ROC curve for discriminating active from inactive EoE patients.

The online version of this article includes the following figure supplement(s) for figure 4:

**Figure supplement 1.** Selective elevation of eosinophil-derived cell-free DNA (cfDNA) in patients with eosinophilic esophagitis (EoE).

cohort, we observed higher levels of B-cell cfDNA in patients (lymphoma n = 10, mean = 1473 GE/ ml; control n = 34, mean = 15 GE/ml, p-value < 0.0001, Mann-Whitney), accompanied by lower B-cell counts and higher T-cell cfDNA (*Figure 5—figure supplement 2*).

These findings indicate that lymphoma growth causes an elevation in the levels of B-cell cfDNA. In addition, a massive loss of normal T-cells leads to extensive release of cfDNA, potentially reflecting an immune response against the tumor or collateral damage. Taken together across all three conditions (influenza vaccination, EoE, and lymphoma), immune cell dynamics in remote locations that are not evident in peripheral blood are detectable via cell-specific methylation markers in plasma.

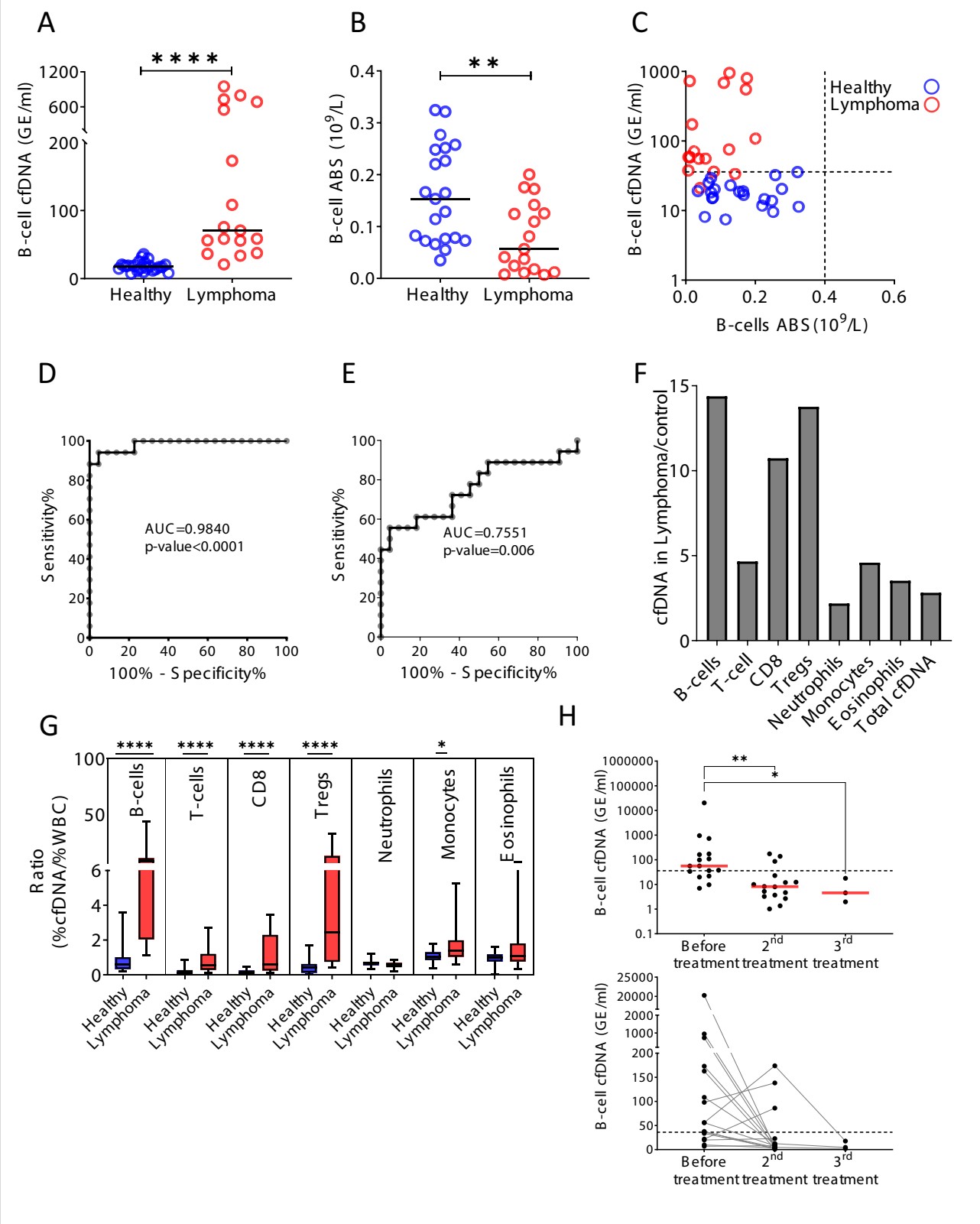

**Figure 5.** B-cell-derived cell-free DNA (cfDNA) elevation in patients with B-cell lymphoma. (**A**) B-cell-derived cfDNA in patients with lymphoma (n = 17) compared with age-matched healthy controls (n = 23, p-value < 0.0001, Mann-Whitney). (**B**) B-cell absolute counts in patients with lymphoma compared with age-matched healthy controls (p-value = 0.0059, Mann-Whitney). (**C**) XY scatter plot for B-cell-derived cfDNA levels versus B-cell absolute counts in blood. Dashed lines indicate healthy baseline levels of B-cell absolute counts in blood and B-cell cfDNA. (**D**) Receiver operating characteristic (ROC)

*Figure 5 continued on next page*

*Figure 5 continued*

curve for the diagnosis of lymphoma based on B-cell cfDNA levels in healthy subjects and patients with B-cell lymphoma. (**E**) ROC curve for diagnosis of lymphoma based on B-cell counts. (**F**) Levels of immune cell type-specific cfDNA in lymphoma patients and healthy controls (mean lymphoma/ mean control). (**G**) The ratio between the percentage of cfDNA from a given immune cell type and the percentage of cells from this population in blood according to complete blood counts (CBC), in each donor among the healthy volunteers (n = 23, blue bars) and patients with lymphoma (n = 17, red bars). Boxes represent 25th and 75th percentiles around the median, whiskers span min to max. (**H**) B-cell-derived cfDNA in lymphoma patients (n = 16) before and after treatment (p-value = 0.0032, Kruskal-Wallis).

The online version of this article includes the following figure supplement(s) for figure 5:

**Figure supplement 1.** Immune-derived cell-free DNA (cfDNA) in lymphoma patients.

**Figure supplement 2.** Cell-free DNA (cfDNA) analysis of a second cohort of lymphoma patients.

**Figure supplement 3.** Cell-free DNA (cfDNA) analysis of a second cohort of lymphoma patients.

## Discussion

We describe a novel method for monitoring turnover dynamics of the human immune system, using cell type-specific cfDNA methylation markers. The assay opens a window into aspects of human immune and inflammation biology that are not reflected in blood cell counts or gene expression patterns. Specifically, the concentration of cfDNA derived from a given immune cell type is a function of the total number of cells of that type (circulating and remote pools, combined), the lifespan of this population, determinants of cfDNA release (e.g. efficiency of phagocytosis) and determinants of cfDNA clearance from plasma (e.g. liver uptake). While many of these parameters are typically unknown, in some cases cfDNA dynamics allow to infer a change in cell turnover or in total cell number outside systemic circulation. For example, when analyzing cfDNA from different cell types in the same sample, it is fair to assume that they were subject to the same clearance kinetics.

We propose that a deeper qualitative and quantitative understanding of the fundamental rules governing cfDNA release and clearance will enrich our ability to relate liquid biopsy data to physiological processes taking place in vivo.

Since the method relies on highly stable methylation marks of cell identity (*Dor and Cedar, 2018*), it is expected to be universal, with the same markers allowing to accurately monitor immune cell dynamics in all individuals. While our current assay uses a panel of 17 methylation markers specific to seven key immune cell types, future improvements should increase the resolution of analysis to target essentially all immune cell types. For example, identification of methylation markers specific to memory B-cells, plasma cells, T-cell subtypes, and tissue-specific macrophages is likely feasible, and could greatly increase the information gained from immune cfDNA analysis. We note however that dynamic cellular states may involve changes in gene expression that do not involve reprogramming of DNA methylation patterns, representing a limitation of the approach. In other words, methylation markers can inform on the turnover dynamics of cell types, not cell states. Analysis of more dynamic aspects of the epigenome, such as the profile of histone marks on circulating nucleosomes (*Sadeh et al., 2021*), may allow inference of transient gene expression programs in cells prior to death and release of cfDNA.

Our cross-sectional and longitudinal analysis of immune cfDNA in healthy individuals begins to define the normal range among the population, an essential step toward using the assay for identifying deviations from health. More extensive characterization of immune cell cfDNA in healthy individuals is necessary to interpret trends that were revealed by our healthy cohorts. For example, we noticed lower levels of neutrophil cfDNA in adult females compared with adult males, suggesting that neutrophils in females live longer; we speculate that such differences in lifespan explain why women have a higher steady-state neutrophil count (*Bain and England, 1975*) (and data not shown). Additional observations of healthy immune cfDNA dynamics that merit further investigation regard age-related changes, such as elevated monocyte cfDNA and reduced lymphocyte cfDNA in individuals older than 60. Finally, the intra- and inter-individual variation in immune cfDNA levels shows that unlike blood cell counts, cfDNA levels vary wildly, apparently with no regulatory mechanism that attracts them to a certain set point typical to an individual. We propose that varying cfDNA levels reflect the action of regulated cell death as a homeostatic mechanism by which the healthy body maintains cell numbers within a desired range (model, *Figure 6*).

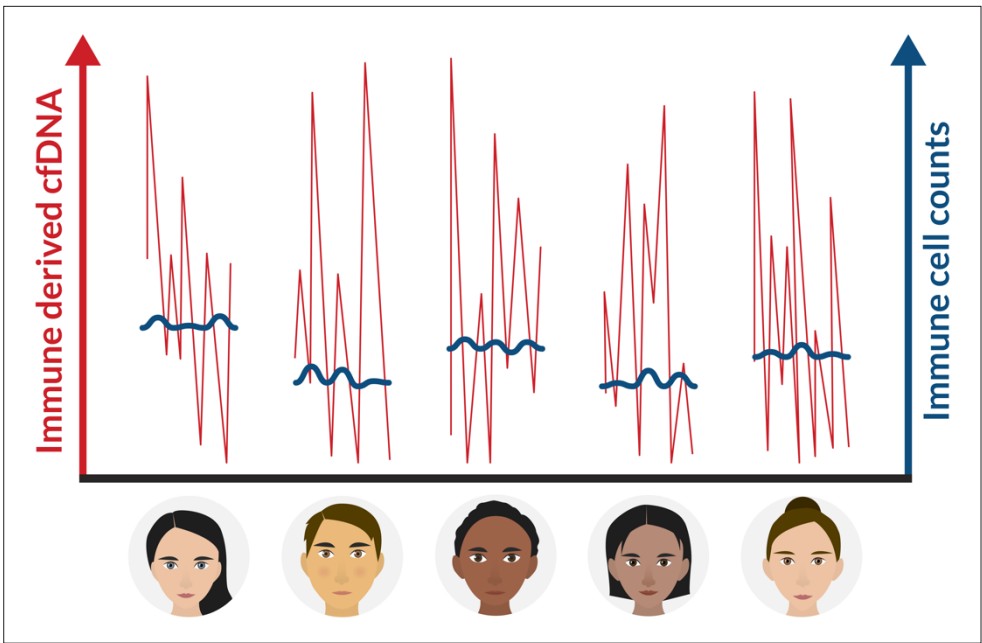

**Figure 6.** A schematic view of immune marker variance within individuals. Intra-individual variance of immune cell count (blue) and immune-derived cell-free DNA (cfDNA) (red) in multiple time points. Our findings suggest that while immune cell counts are stable and typical within an individual, immune cell cfDNA levels vary greatly, reflecting changes in cell turnover that help maintain the cell count set point.

This model has some practical implications. For diagnostic applications of immune cfDNA, one should use the healthy population baseline levels of each methylation marker. Deviation from the normal range would indicate abnormality that requires attention (similar to the situation with blood counts). Our study suggests that fluctuations of specific cfDNA markers within norm levels are not indicative of pathology, but are rather part of a normal homeostatic process.

Beyond the healthy baseline, we studied immune cfDNA dynamics in three settings of immune system perturbation. First, post influenza vaccination, we identified an early elevation of B-cell cfDNA, preceding an increase in circulating B-cell counts and correlating to the effectiveness of antibody production. The correlation was statistically significant and independent of the known age-related risk of non-responsiveness. We propose that elevated B-cell turnover (and elevated B-cell cfDNA as its readout) reflects early stages in the successful response of B-cells to the vaccine, including the process of affinity maturation whereby large numbers of B-cells are generated and eliminated within lymph nodes as a result of insufficient binding to the target epitope. More work is needed to accurately define the population of B-cells that release cfDNA after vaccination, and to understand the physiological driver of this response. Practical applications, if the association between B-cell cfDNA and vaccination outcome proves robust, may include an early personalized indication for the success of vaccination. Second, we examined immune cfDNA dynamics in EoE, a model for an inflammatory disease in which one tissue is damaged by infiltration of a specific immune cell population, while leaving a minimal mark on peripheral cell counts. cfDNA analysis revealed the selective elevation of eosinophil turnover in active EoE, in some cases even when circulating eosinophil cell counts are unchanged. Larger scale studies are warranted to determine if eosinophil cfDNA can be a sufficiently sensitive and specific biomarker for assisting the clinical diagnosis and monitoring of EoE, ultimately relieving the need for invasive biopsies of the esophagus. Lastly, cfDNA dynamics in patients with B-cell lymphoma revealed the impact of disease on the turnover of B-cells (consistent with a previous study by *Sun et al., 2015*), and the reflection in B-cell cfDNA of the response to treatment. This analysis also revealed the involvement of turnover of other immune cell types in lymphoma. As with EoE, cfDNA in lymphoma provides a systemic biomarker of immune processes taking place in remote locations. However in lymphoma, these processes include both tumor dynamics and host responses – either bystander effects (collateral damage) or an immune response to the tumor. Potential uses of immune methylation markers in this field include early diagnosis of hematological malignancies, detection of minimal residual disease,

and monitoring response to treatment (although our findings suggest that the latter may involve complex relationships between B-cell cfDNA and clinical response). Beyond hematological malignancies, immune-derived cfDNA dynamics can inform on the response to immune checkpoint inhibitors.

It is important to put the novel assay of immune cfDNA in the context of other emerging immune monitoring tools. In conditions that involve activation of the adaptive immune system, sequencing of the B- and T-cell receptor repertoire can be highly informative regarding the nature of the involved B- and T-cell clones, and the nature of the specific target epitopes. An important current limitation of immune repertoire analysis is that it requires large genomic fragments, which can only be derived from whole cells. This precludes cfDNA-based analysis, and limits access to T- and B-cell dynamics taking place in remote locations. Expression profiling of leukocytes, using single cell RNA sequencing and CyTOF, is also emerging as a powerful tool for understanding immune cell dynamics, revealing rich information about the transcriptome of circulating cells (*Bucasas et al., 2011*; *Jiang et al., 2013*; *Kurtz et al., 2015*; *Voigt et al., 2018*; *Meng et al., 2017*). We propose that both T/B repertoire analysis and expression profiling can be complemented by cfDNA profiling, which will provide non-overlapping information about immune cell turnover (and potentially also gene expression patterns associated with cell death) (*Sadeh et al., 2021*) in remote locations.

In summary, analysis of specific immune cell methylation markers in cfDNA allows for monitoring of human immune cell dynamics, providing temporal and spatial information not accessible via circulating cell counts. We propose that this novel tool can illuminate healthy and pathological immune processes, including non-immune diseases having an inflammatory component such as cancer, rejection of transplanted organs, metabolic and neurodegenerative disease.

## Materials and methods
### Subject enrollment
This study was conducted according to protocols approved by the Institutional Review Board at each study site, with procedures performed in accordance with the Declaration of Helsinki. Blood and tissue samples were obtained from donors who have provided written informed consent. When using material from deceased organ donor those with legal authority were consented. Subject characteristics are presented in *Supplementary file 1*.

### Healthy controls
A total of 234 healthy volunteers (56% females, 44% males, age range 1–85 years) participated in the study as unpaid healthy controls. All denied having any chronic or acute disease.

### Temporal experiment cohort
Fifteen healthy volunteers gave blood each week for 6 weeks (9 females, 6 males, age 21-68 years).

### Vaccination cohort and determination of anti-hemagglutinin antibody titers
Ninety-two healthy volunteers that received the annual influenza vaccination (55 females, 37 males, age range 20-73 years) gave blood samples a day before vaccination, and after 3, 7, and 28 days (±2 days).

The anti-hemagglutinin antibody titers were determined using hemagglutination inhibition (HI) assay. Serum samples obtained from vaccinated and non-vaccinated individuals were stored at −20°C until tested treated with receptor destroying enzyme (RDE) (Sigma C8772, diluted 1:4), for 16 hr prior to heat inactivation (30 min, 56°C). Absorption with erythrocytes was performed to remove non-specific hemagglutination, in accordance with a modified WHO protocol (*Rowe et al., 1999*). Serial 2-fold dilutions (1:20–1:2560) of sera in 25 µl PBS were prepared in V-shaped well plates, and an equal volume of four hemagglutinin units of viral antigen was added. The mixture was then incubated at room temperature for 1 hr. Fifty microliters of 0.5% chicken erythrocytes suspended in PBS, were added to the wells, and mixed by shaking the plates on a mechanical vibrator. Agglutination patterns were read after 30 min and the HI titer was defined as the reciprocal of the last dilution of serum that fully inhibited hemagglutination. The cut-off value selected for a positive result was 1:40. The influenza antigens for 2018–19 and 2019–20 winter seasons were supplied by the WHO.

Responders were defined as people who did not have at baseline antibodies against at least one of the four strains (2018–19: H1N1, H80, YAMA, VIC; 2019–20: H1N1, YAMA, VIC, H3; no antibodies defined as influenza strain antigen titer <40) and developed antibodies after vaccination (influenza strain antigen titer >40). One person had antibodies against all strains at baseline and was excluded.

### EoE cohort

Twenty-one active EoE patients, 24 EoE patients in remission, and 14 controls were recruited to the study at Cincinnati Children's Hospital. Diagnosis of EoE patients was made based on an histological biopsy taken from the distal esophageal tissue.

### Lymphoma cohort

Twenty-seven newly diagnosed lymphoma patients that came for treatment in the hematological daycare unit in Hadassah Medical Center were recruited to the study in two cohorts (17 patients in cohort #1, 10 patients in cohort #2). Diagnosis was made by PET-CT. In addition, we recruited 16 newly diagnosed lymphoma patients who were monitored before and after receiving treatment. Response to treatment was assessed based on PET-CT.

### Sample collection and processing

Blood samples were collected by routine venipuncture in 10 ml EDTA Vacutainer tubes or Streck blood collection tubes and stored at room temperature for up to 4 hr or 5 days, respectively. Tubes were centrifuged at 1500× $g$ for 10 min at 4°C (EDTA tubes) or at room temperature (Streck tubes). The supernatant was transferred to a fresh 15 ml conical tube without disturbing the cellular layer and centrifuged again for 10 min at 3000× $g$. The supernatant was collected and stored at –80°C. We note that immune cfDNA analysis is particularly sensitive to conditions of plasma isolation, given the potential confounding effects of DNA released from lysed leukocytes. Our experiments indicated that the isolation protocol described above is optimal. While the speed of second centrifugation (3000 × $g$ or higher) did not have a major effect of yield and purity, consistent with previous work (*Risberg et al., 2018*; *Ungerer et al., 2020*), it was important to minimize the time spent between blood drawing and centrifugation when using EDTA tubes. cfDNA was extracted from 2 to 4 ml of plasma using the QIAsymphony liquid handling robot (Qiagen). cfDNA concentration was determined using Qubit double-strand molecular probes kit (Invitrogen) according to the manufacturer's instructions.

DNA derived from all samples was treated with bisulfite using EZ DNA Methylation-Gold (Zymo Research), according to the manufacturer's instructions, and eluted in 24 μl elution buffer.

### Immune cell and tissue isolation and processing

PBMCs from a healthy individual were isolated using ficoll-paque density gradient (Miltenyi Biotec). CD4+ T-cells, CD8+ T-cells, CD19+ B-cells, and Nk CD56+ cells were positively selected using magnetic MicroBeads. Monocytes were negatively selected (Miltenyi Biotec) as instructed by the manufacturer. Tregs (CD4+, CD25+, FOXP3+, 28.5% purity) were purchased from Astarte biologics. Neutrophils and eosinophils were isolated based on a previously published protocol (*Hartman et al., 2001*; *Sagiv et al., 2016*). Genomic DNA from other tissues was purchased as previously described (*Lehmann-Werman et al., 2016*; *Zemmour et al., 2018*).

### Selection of immune cell methylation markers

Immune cell-specific methylation candidate biomarkers were selected using comparative methylome analysis, based on publicly available datasets (*Moss et al., 2020*; *Moss et al., 2018*), to identify loci having more than five CpG sites within 150 bp, with an average methylation value for a specific cytosine (present on Illumina 450K arrays) of less than 0.3 in the specific immune cell type and greater than 0.8 in over 90% of tissues and other immune cells. As noted above, such putative marker loci are far more abundant in the genome than loci that are methylated in the cell type of interest and unmethylated elsewhere. From our previously described atlas of human tissue-specific methylomes (*Lehmann-Werman et al., 2016*; *Moss et al., 2018*), we identified ~50 CpG sites that are unmethylated in specific immune cells and methylated in all other major immune cells and tissues. We selected arbitrarily two to three of these sites for neutrophils (i.e. NEUT1, NEUT2, NEUT3), monocytes (i.e. MONO1, MONO2), eosinophils (i.e. EOSI1, EOSI2, EOSI3), B-cells (i.e. B-CELL1, B-CELL2), T-cells

(i.e. T-CELL1, T-CELL2), CD8 T-cells (CD8A, CD8B), Tregs (TREG1, TREG2), and designed primers to amplify ~100 bp fragments surrounding them using the multiplex two-step PCR amplification method (*Neiman et al., 2020*). Marker coordinates and primer sequences are provided in *Supplementary file 2*.

The validation of markers was done using DNA extracted from different cells and tissues, and the methylation status of the CpG block was assessed. Some markers were more sensitive if one CpG site was allowed to be methylated differently than other CpGs in the block. as indicated in *Supplementary file 2*.

## PCR

To efficiently amplify and sequence multiple targets from bisulfite-treated cfDNA, we used a two-step multiplexed PCR protocol, as described recently (*Neiman et al., 2020*). In the first step, up to 17 primer pairs were used in one PCR reaction to amplify regions of interest from bisulfite-treated DNA, independent of methylation status. Primers were 18–30 base pairs (bp) with primer melting temperature ranging from 58°C to 62°C. To maximize amplification efficiency and minimize primer interference, the primers were designed with additional 25 bp adaptors comprising Illumina TruSeq Universal Adaptors without index tags. All primers were mixed in the same reaction tube. For each sample, the PCR was prepared using the QIAGEN Multiplex PCR Kit according to manufacturer's instructions with 7 µl of bisulfite-treated cfDNA. Reaction conditions for the first round of PCR were: 95°C for 15 min, followed by 30 cycles of 95°C for 30 s, 57°C for 3 min and 72°C for 1.5 min, followed by 10 min at 68°C.

In the second PCR step, the products of the first PCR were treated with Exonuclease I (ThermoScientific) for primer removal according to the manufacturer's instructions. Cleaned PCR products were amplified using one unique TruSeq Universal Adaptor primer pair per sample to add a unique index barcode to enable sample pooling for multiplex Illumina sequencing. The PCR was prepared using 2× PCRBIO HS Taq Mix Red Kit (PCR Biosystems) according to manufacturer's instructions. Reaction conditions for the second round of PCR were: 95°C for 2 min, followed by 15 cycles of 95°C for 30 s, 59°C for 1.5 min, 72°C for 30 s, followed by 10 min at 72°C. The PCR products were then pooled, run on 3% agarose gels with ethidium bromide staining, and extracted by Zymo GEL Recovery kit.

## NGS and analysis

Pooled PCR products were subjected to multiplex NGS using the MiSeq Reagent Kit v2 (Illumina) or the *NextSeq* 500/550 v2 Reagent Kit (Illumina). Sequenced reads were separated by barcode, aligned to the target sequence, and analyzed using custom scripts written and implemented in R. Reads were quality filtered based on Illumina quality scores. Reads were identified as having at least 80% similarity to the target sequences and containing all the expected CpGs. CpGs were considered methylated if 'CG' was read and unmethylated if 'TG' was read. Proper bisulfite conversion was assessed by analyzing methylation of non-CpG cytosines. We then determined the fraction of molecules in which all CpG sites were unmethylated. The fraction obtained was multiplied by the concentration of cfDNA measured in each sample, to obtain the concentration of tissue-specific cfDNA from each donor. Given that the mass of a haploid human genome is 3.3 pg, the concentration of cfDNA could be converted from units of ng/ml to haploid GE/ml by multiplying by a factor of 303.

Methylation markers were calibrated compared to CBC to give a more accurate quantitative number, using linear regression. When necessary, methylation values were multiplied by a coefficient number derived from our spike-in calibration curves, to reflect the actual concentration of DNA from the relevant cell type. The fraction of neutrophil-derived cfDNA was obtained by dividing methylation fraction by 0.69.

Our entire dataset of PCR sequencing reactions used in this study is available upon request. The computational pipeline used to interpret sequence reads as well as a representative set of data (sequences that gave rise to *Figure 1C*) were uploaded to GitHub (https://github.com/Joshmoss11/btseq; swh:1:rev:efc75ddd347c20392cf0a034706a7b5b6090be75, *Moss, 2021*) .

## Deconvolution

We obtained 46 WGBS datasets from 23 healthy adult individuals. For each of these donors, we extracted genomic DNA from white blood cells (WBC) and cfDNA from plasma, and performed WGBS

at an average depth of 85×. Methylation data was uploaded to GEO (Accession number GSE186888). WGBS data were converted to an array-like format by calculating the average methylation at 7890 CpGs from the Moss et al. methylation atlas (*Moss et al., 2018*). We then ran the deconvolution algorithm (*Loyfer, 2018*, https://github.com/nloyfer/meth_atlas) for each WBC and cfDNA sample, to assess the relative presence of each blood cell type.

## Statistics

To assess the correlation between groups, we used Pearson's correlation test. To determine the significance of differences between groups we used a non-parametric two-tailed Mann-Whitney test. For multiple comparisons, a Kruskal-Wallis multiple comparison test was used. p-Value was considered significant when < 0.05. To detect outliers in the healthy population we applied a multiple outlier detection ROUT-test (Q = 5%) (*Motulsky and Brown, 2006*). Samples that were detected as outliers were excluded. All statistical analyses were performed with GraphPad Prism 8.4.3.

## Intra-individual and inter-individual variation

Intra-individual coefficient of variation for each immune cell type in CBC, whole blood, and cfDNA was calculated for each person across six different time points. The inter-individual coefficient of variation for each immune cell type was calculated for each time point across all individuals. The average of the intra-individual coefficient of variation was calculated. To prevent a bias due to difference in sample size (intra-individual variation, six time points; inter-individual variation, 15 individuals), we used R (version3.6.1) to sample all different combinations of a randomly selected six-person group and calculated the inter-individual coefficient of variation. Coefficients of variation of the different combinations were averaged.

## Acknowledgements

We thank Daniela Beller from the Maccabi TIPA biobank for providing data on blood cell counts, and the many volunteers who donated blood for this study. We thank Shai Shen-Orr for critical reading of the manuscript, and Nir Friedman, Ron Milo, Ron Sender, and Mordechai Slae for fruitful discussions. We thank Idit Shiff and Abed Nasseredin from the Genomics lab of the Core Research Facility (CRF) at The Faculty of Medicine, The Hebrew University of Jerusalem for their support in DNA and sequencing analysis.

This work was supported by grants from Human Islet Research Network (HIRN UC4DK116274 and UC4DK104216 to YuD); JDRF (3-SRA-2014–38 and 1-SRA-2019–705), Ernest and Bonnie Beutler Research Program of Excellence in Genomic Medicine, The Alex U Soyka pancreatic cancer fund, The Israel Science Foundation, the Waldholtz / Pakula family, the Robert M and Marilyn Sternberg Family Charitable Foundation, the Helmsley Charitable Trust, Grail, and the DON Foundation (to YuD). YuD holds the Walter and Greta Stiel Chair and Research grant in Heart studies. IF-F received a fellowship from the Glassman Hebrew University Diabetes Center.

## Additional information

### Competing interests

Ilana Fox-Fisher, Judith Magenheim, Joshua Moss, Tommy Kaplan, Benjamin Glaser, Ruth Shemer, Yuval Dor: has filed patents on cfDNA analysis technology. Arash Jamshidi, Gordon Cann: is an employee of GRAIL. The other authors declare that no competing interests exist.

### Funding

| Funder | Grant reference number | Author |
| --- | --- | --- |
| Human Islet Research Network | HIRN UC4DK116274 | Ruth Shemer |
| Human Islet Research Network | HIRN UC4DK104216 | Yaron Drori |

| Funder | Grant reference number | Author |
|---|---|---|
| Juvenile Diabetes Research Foundation, International | 3-SRA-2014–38 | Yuval Dor |
| Juvenile Diabetes Research Foundation, International | 1-SRA-2019–705 | Yuval Dor |
| Ernest and Bonnie Beutler Research Program of Excellence in Genomic Medicine | | Yuval Dor |
| Alex U Soyka Pancreatic Cancer Research Fund | | Yuval Dor |
| Israel Science Foundation | | Yuval Dor |
| Waldholtz/Pakula family | | Yuval Dor |
| Robert M and Marilyn Sternberg Family Charitable Foundation | | Yuval Dor |
| Helmsley Charitable Trust | | Yuval Dor |
| GRAIL | | Yuval Dor |
| The DON Foundation | | Yuval Dor |
| Walter and Greta Stiel | Chair | Yuval Dor |
| Glassman Hebrew University Diabetes Center | | Ilana Fox-Fisher |
| Walter and Greta Stiel | Research grant in heart studies | Yuval Dor |

No external funding was received for this work.

## Author contributions

Ilana Fox-Fisher, Conceptualization, Investigation; Sheina Piyanzin, Bracha Lea Ochana, Agnes Klochendler, Judith Magenheim, Ayelet Peretz, Yaron Drori, Nehemya Friedman, Michal Mandelboim, Investigation; Netanel Loyfer, Tommy Kaplan, Formal analysis; Joshua Moss, Data curation, Software; Daniel Cohen, Data curation, Formal analysis; Marc E Rothenberg, Julie M Caldwell, Mark Rochman, Arash Jamshidi, Gordon Cann, David Lavi, Resources; Benjamin Glaser, Ruth Shemer, Conceptualization, Supervision; Yuval Dor, Conceptualization, Funding acquisition, Supervision, Writing – original draft

## Author ORCIDs

Ilana Fox-Fisher http://orcid.org/0000-0002-3773-9330
Mark Rochman http://orcid.org/0000-0002-0818-927X
Tommy Kaplan http://orcid.org/0000-0002-1892-5461
Yuval Dor http://orcid.org/0000-0003-2456-2289

## Ethics

Human subjects: This study was conducted according to protocols approved by the Institutional Review Board at each study site (Hadassah Medical Center: HMO-14-0198. A Method to Diagnose Cell Death Based on Methylation Signature of Circulating Cell-Free DNA, Cininnati Children's Hospital: CCHMC IRB protocol 2008-0090: Eosinophils and Inflammation, an Expanded Study), with procedures performed in accordance with the Declaration of Helsinki. Blood and tissue samples were obtained from donors who have provided written informed consent. When using material from deceased organ donor those with legal authority were consented. Subject characteristics are presented in Supplementary file 1.

## Decision letter and Author response

Decision letter https://doi.org/10.7554/eLife.70520.sa1
Author response https://doi.org/10.7554/eLife.70520.sa2

## Additional files

### Supplementary files
- Supplementary file 1. Blood donor characteristics.
- Supplementary file 2. Genomic coordinates of immune cell type-specific methylation markers used in this study, and primer sequences used to amplify these loci after bisulfite conversion.
- Transparent reporting form

### Data availability
All data generated or analyzed during this study are included in the manuscript and supporting files. The whole-genome bisulfite sequencing data reported in the paper, from 46 samples, is uploaded to GEO as described. The paper also reports data from PCR reactions that were analyzed by massively parallel sequencing. This is a very large set of data that is extremely low in information content and is of little interest to readers or even to people interested in replicating our results or interrogating them further. The key information (methylation status) in each sample is provided in the supplementary information, and we also uploaded the analysis algorithm and some sequence data. The entire set of raw sequencing data is available in the Dor lab to anyone interested. Please contact Prof. Yuval Dor dor@huji.ac.il. All information will be shared. There is no need for any paperwork. Code is uploaded to GitHub as described in the paper. The methylation status of each marker in each sample is provided in Supplementary file 1. This data was used to generate the graphs shown in the paper. Sheets in this file indicate which figure they relate to.

The following dataset was generated:

| Author(s) | Year | Dataset title | Dataset URL | Database and Identifier |
|---|---|---|---|---|
| Fox-Fisher I, Piyanzin S, Ochana B, Klochendler A, Magenheim J, Peretz A, Loyfer N, Moss J, Cohen D, Drori Y, Friedman N, Mandelboim M, Rothenberg ME, Caldwell JM, Rochman M, Jamshidi A, Cann G, Lavi D, Kaplan T, Glaser B, Shemer R, Dor Y | 2021 | Remote immune processes revealed by immune-derived circulating cell-free DNA | https://www.ncbi.nlm.nih.gov/geo/query/acc.cgi?acc=GSE186888 | NCBI Gene Expression Omnibus, GSE186888 |

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
