## [Decision Letter]

**Decision letter after peer review:**

Thank you for submitting your article "Remote immune processes revealed by immune-derived circulating cell-free DNA" for consideration by *eLife*. Your article has been reviewed by 2 peer reviewers, one of whom is a member of our board of Reviewing Editors, and the evaluation has been overseen by Y M Dennis Lo as the Senior Editor. The reviewers have opted to remain anonymous.

Essential revisions:

1) Both reviewers indicated that the primary data and analysis routines are not accessible. Please share all data and analysis routines in a public repository.

2) Please address the question raised by R3 regarding the possibility/need to distinguish B cell subtypes with new experiments, new analysis or, alternatively, additional discussion.

*Reviewer #1 (Recommendations for the authors):*

This is an interesting manuscript in which Fox-Fisher et al., report a novel method for minimally invasive monitoring of the dynamics of the human immune system, based on profiling of cell-type specific cell-free DNA methylation marks. This method implements targeted DNA sequencing of a small panel of cell-type specific CpG markers and has the potential to be useful for the monitoring of a wide range immune-related diseases. The authors have tested the utility of this assay in three settings: influenza vaccination, Eosinophilic Esophagitis and B cell lymphoma. The experiments are well described and well powered, and the results are compelling, and I am happy to recommend this paper for publication in *eLife* after the following points are addressed:

1. I appreciate that the authors shared PCR primer sequences. To ensure reproducibility it is important that all sequencing data, as well as analysis scripts or software are shared in a public resource.

2. The presentation of the data in some of the figures can be improved. The cell type labels in panels A, B and C are not consistent. Please indicate the number of samples evaluated in all box, bar and scatter plots (Figures 2D, 3A-C, 3E-F, 4A-C, 5A-B, 5G-H).

3. The observation of significant immune cfDNA dynamics and intra- and interindividual variation is interesting, and the schematic of Figure 6 is helpful. Some discussion of the challenges that this variability brings along for diagnostic applications would be helpful, and it would be great if the authors can suggest solutions.

4. The statistical analysis of the dynamics of B-cell cfDNA for vaccine responders vs non-responders can be improved. How were responders and non-responders selected and how sensitive is the cfDNA analysis to the selection criteria used? The differences observed for the two groups are relatively minor (certainly not striking as claimed in the discussion, please tone this down), and may be largely due to small number of samples from responders that are analyzed?

5. The ROC analysis of the performance of eosinophil cfDNA to discriminate patients with active EoE and healthy controls is helpful. Please provide a similar analysis of the performance of your assay to discriminate inactive EoE from no disease, and to discriminate active and inactive EoE.

6. The lymphoma patients are newly diagnosed and treatment naïve, which is appropriate because treatment would likely lead to a greatly increased burden of B cell cfDNA. Do you have access to samples collected during treatment? It would be interesting to evaluate the possibility to monitor treatment responses and disease recurrence.

7. I would like to hear your thoughts (in the Discussion section), for ways to disentangle the different parameters that define the amount of cfDNA from specific immune cell types in blood (turnover rate, total number of cells, determinants of clearance and release).

8. The Discussion section would benefit from broader and more deeper discussion of other immune monitoring tools that have been reported or that are used in clinical practice, and the relative merit of the immune cfDNA assay the authors describe. For example, immune repertoire analyses of B and T cell compartments have been used to evaluate vaccine response (see the work by Jiang and Quake, and Vollmers and Quake, for example), and to monitor lymphoma. Other assays to monitor inflammation marks, or that use gene expression profiling are in use, or have been used.

*Reviewer #2 (Recommendations for the authors):*

1. The authors should investigate the subtypes of B cells and redo the analysis. In addition As reported by the authors and others (Lam et al., Clin Chem 2017, Moss et al., Nat Commun 2018), blood cells of erythroid lineage also contribute quite a lot in cfDNA, therefore the authors need to validate the specificity of their CpG loci in erythroblasts.

2. The authors clearly stated that they only work on hypo-methylated CpGs, why? I understand that biologically there may not be tissue-specific hyper-methylated CpGs, but the author need to explain this point.

3. Elevation of cfDNA from B-cells in lymphoma has been reported before (Sun et al., PNAS 2015), the authors need to mention such information.

4. Figure 1B is very unclear, what do the values represent? Do they denote the proportions in the mixture?

5. The presentation of Figure 1C is not easy to follow. A barplot with cumulative frequency for all cell types could be better.

6. The authors miss citations of many important prior works in their citation. This needs to be improved in a revision.

7. The blood processing procedure is not as commonly used, especially that the author uses only 3,000g for the 2^nd^ step of centrifusion, while the common approach uses ~16,000g, so I wonder is there any contaminations in cfDNA? Which could significantly affect the result as this work specifically look at immune cells.

8. The data could only partially support the aims and conclusions, and the work is in principle-of-concept stage that needs further enhancements and validations.

9. I could not find any statement on data release in the article. Please kindly add this in a revised manuscript.

---

## [Author Response]

Essential revisions:1) Both reviewers indicated that the primary data and analysis routines are not accessible. Please share all data and analysis routines in a public repository.

Thank you for the positive evaluation of the paper.

We have now more fully explained in the methods section the experimental protocols, both wet and computational. These include:

– Full primer sequences (Supplementary Table S1)

– A more detailed protocol for our multiplex PCR, including a reference to our recently published protocol (Methods).

– A more detailed description of analysis routines i.e. how primary sequences are interpreted. This data was uploaded to GitHub.

– A more detailed description of our WGBS deconvolution (Methods).

With regard to primary data that was not provided in the first version: we described a targeted assay, which generates a large number of sequence reads from the same locus, differing in their methylation patterns. The information content of each sequence run is pretty thin, and can be summarized as the fraction of molecules that carry a specific pattern of methylation (which can identify the cell type of origin). In the course of the study we performed many thousands of PCR reactions (hundreds of plasma samples, 17 markers in each), and each product was sequenced >10,000 times. We are happy to share the entire set of raw data with anyone interested, but do not see the value of posting this information online. In the revised version we provide the algorithm used to interpret the raw sequence data, and a sample of sequence output using multiplex PCR followed by sequencing, on the genomic DNA samples described in Figure 1C (uploaded to GitHub). This example will make interested readers understand how the analysis is performed.

We have also shared 46 WGBS methylomes of healthy controls (white blood cells and plasma), which were deconvoluted to provide an independent and unbiased view of the contribution of different immune cell types to healthy cfDNA (Figure 2D). Data was uploaded to GEO.

2) Please address the question raised by R3 regarding the possibility/need to distinguish B cell subtypes with new experiments, new analysis or, alternatively, additional discussion.

Developing new methylation biomarkers for additional immune cell types is a major undertaking that falls beyond the scope of this paper, even though we recognize its importance. In the revision we discuss the topic in more detail – the need for additional cell type-specific markers, the potential and the limitations (Discussion, 2^nd^ paragraph).

We do note that we have added new experiments in the revised version, using the same panel of methylation markers, to examine another aspect of tissue dynamics in lymphoma patients (response to treatment), as requested by Reviewer 2.

Reviewer #1 (Recommendations for the authors):This is an interesting manuscript in which Fox-Fisher et al., report a novel method for minimally invasive monitoring of the dynamics of the human immune system, based on profiling of cell-type specific cell-free DNA methylation marks. This method implements targeted DNA sequencing of a small panel of cell-type specific CpG markers and has the potential to be useful for the monitoring of a wide range immune-related diseases. The authors have tested the utility of this assay in three settings: influenza vaccination, Eosinophilic Esophagitis and B cell lymphoma. The experiments are well described and well powered, and the results are compelling, and I am happy to recommend this paper for publication in eLife after the following points are addressed:1. I appreciate that the authors shared PCR primer sequences. To ensure reproducibility it is important that all sequencing data, as well as analysis scripts or software are shared in a public resource.

Revision contains a more detailed description of methods, including the protocol for multiplex PCR (Methods), primer sequences (Supplementary Table S1) and analysis routines (uploaded to GitHub).

Note that the assay described in this paper is targeted, such that each PCR reaction generated ~10,000 reads of the same sequence, differing in methylation status which indicates the cell of origin. We are happy to share with any interested investigator the many thousands of sequence datasets that gave rise to the results presented in the paper, however we think this will not be useful. In a sense, a reproducibility test of our findings will be much easier achieved by simply ordering a few pairs of primers and running PCR-sequencing reaction on fresh cfDNA samples! However, to demonstrate the analytic process fully, we uploaded to GitHub a representative sequencing dataset (Figure 1C) in its raw form, as well as a demonstration of its analysis.

2. The presentation of the data in some of the figures can be improved. The cell type labels in panels A, B and C are not consistent.

We assume that this comment refers to Figure 1. We revised the labels to be consistent throughout the paper.

Please indicate the number of samples evaluated in all box, bar and scatter plots (Figures 2D, 3A-C, 3E-F, 4A-C, 5A-B, 5G-H).

The number of samples is now indicated for each panel. For clarity of visualization we prefer to provide this information in the figure legends rather than in the panels.

3. The observation of significant immune cfDNA dynamics and intra- and interindividual variation is interesting, and the schematic of Figure 6 is helpful. Some discussion of the challenges that this variability brings along for diagnostic applications would be helpful, and it would be great if the authors can suggest solutions.

Thank you. The issue is now better discussed (3^rd^ paragraph in the revised Discussion section).

4. The statistical analysis of the dynamics of B-cell cfDNA for vaccine responders vs non-responders can be improved. How were responders and non-responders selected and how sensitive is the cfDNA analysis to the selection criteria used? The differences observed for the two groups are relatively minor (certainly not striking as claimed in the discussion, please tone this down), and may be largely due to small number of samples from responders that are analyzed?

The definitions of responders and non-responders are now better explained in the methods section. Briefly, definition is based on the levels of antibodies observed. Responders are people who did not have antibodies (any of the 4 measured) prior to vaccination, and showed a titer >40 when measured 28 days after vaccination. Non-responders are individuals who did not show elevation in any of the antibodies after vaccination.

We added now a new panel (Figure 3F) showing the ROC curve for discrimination of responders from non-responders based on peak B-cell cfDNA post-vaccination. The signal is statistically significant so should not be ignored, but is indeed not striking. As requested, we toned down the statement regarding prediction of response (Discussion).

5. The ROC analysis of the performance of eosinophil cfDNA to discriminate patients with active EoE and healthy controls is helpful. Please provide a similar analysis of the performance of your assay to discriminate inactive EoE from no disease, and to discriminate active and inactive EoE.

Thank you for this excellent suggestion. We found that indeed eosinophil cfDNA can discriminate active from inactive EoE. We added a new panel (4F) showing a ROC curve to this end.

6. The lymphoma patients are newly diagnosed and treatment naïve, which is appropriate because treatment would likely lead to a greatly increased burden of B cell cfDNA. Do you have access to samples collected during treatment? It would be interesting to evaluate the possibility to monitor treatment responses and disease recurrence.

Thank you, we had access to samples from patients who received treatment (n=16) and added information on their B-cell cfDNA levels to figure 5 (new panel 5H). The results show an overall decrease in B-cell derive cfDNA after treatment, as might have been expected.

7. I would like to hear your thoughts (in the Discussion section), for ways to disentangle the different parameters that define the amount of cfDNA from specific immune cell types in blood (turnover rate, total number of cells, determinants of clearance and release).

Thank you. We elaborate on this important issue in the revised Discussion. In brief, we stated that it is possible to extract important information from cfDNA analysis even without knowing much about cfDNA release and clearance (as practice shows). However we propose that a qualitative and quantitative understanding the fundamental rules governing cfDNA dynamics will enrich our ability to relate cfDNA data to physiological processes taking place in vivo.

8. The Discussion section would benefit from broader and more deeper discussion of other immune monitoring tools that have been reported or that are used in clinical practice, and the relative merit of the immune cfDNA assay the authors describe. For example, immune repertoire analyses of B and T cell compartments have been used to evaluate vaccine response (see the work by Jiang and Quake, and Vollmers and Quake, for example), and to monitor lymphoma. Other assays to monitor inflammation marks, or that use gene expression profiling are in use, or have been used.

Thank you. We have added a paragraph to the Discussion about cfDNA in the context of other immune monitoring tools.

Reviewer #2 (Recommendations for the authors):1. The authors should investigate the subtypes of B cells and redo the analysis. In addition As reported by the authors and others (Lam et al., Clin Chem 2017, Moss et al., Nat Commun 2018), blood cells of erythroid lineage also contribute quite a lot in cfDNA, therefore the authors need to validate the specificity of their CpG loci in erythroblasts.

Thank you. Indeed, the next step of this work should be increasing the resolution of methylation markers to include additional cell types, for example memory B cells and plasma cells, memory and effector T cell, and tissue-specific macrophages. However this is a major undertaking that falls beyond the scope of this work. In the revised discussion we describe the potential and the limitations of expansion of the spectrum of cellular subtypes that can be tested in cfDNA. With regard to erythroblasts, this is indeed an important cell type to include in specificity analysis of methylation markers. This is now included in the revised heatmap in Figure 1B. All markers but one retained their specificity.

2. The authors clearly stated that they only work on hypo-methylated CpGs, why? I understand that biologically there may not be tissue-specific hyper-methylated CpGs, but the author need to explain this point.

Indeed, the majority of our cell type-specific markers (in the immune system and beyond) are hypo-methylated in the cell type of interest and hypermethylated elsewhere. Occasionally we do identify loci that are hypermethylated specifically in the cell type of interest, and these work just as fine. The reason for the more frequent use of hypomethylated markers is that these are far more abundant in the genome, typically reflecting tissue-specific gene enhancers. This is clarified in the revised manuscript (first section in Results).

3. Elevation of cfDNA from B-cells in lymphoma has been reported before (Sun et al., PNAS 2015), the authors need to mention such information.

Thank you. Sun et al., was unintentionally not mentioned. It is now cited in the Results (lymphoma section) and Discussion.

4. Figure 1B is very unclear, what do the values represent? Do they denote the proportions in the mixture?

Thank you. We changed the presentation of the panel as well as the legend to clarify the meaning of the shades of grey. Grey blocks represent the percentage of unmethylated molecules of each marker (X axis) in DNA from each cell type (Y axis).

5. The presentation of Figure 1C is not easy to follow. A barplot with cumulative frequency for all cell types could be better.

Supplementary Figure S1 provides the information from the spike-in experiment for each marker, in the form of a bar plot as requested. We did generate a cumulative bar plot but noticed that is more difficult to understand than the version shown in Figure 1C. Text was revised to clarify this plot.

6. The authors miss citations of many important prior works in their citation. This needs to be improved in a revision.

Thank you. We added key references that were unfortunately omitted in the first version.

7. The blood processing procedure is not as commonly used, especially that the author uses only 3,000g for the 2nd step of centrifusion, while the common approach uses ~16,000g, so I wonder is there any contaminations in cfDNA? which could significantly affect the result as this work specifically look at immune cells.

Our blood processing procedure is similar to that used by others in the field. In fact, different centrifugation procedures have been compared systematically and found to be equivalent. The revised Methods section cites papers that performed the comparison (Effects of Collection and Processing Procedures on Plasma Circulating Cell-Free DNA from Cancer Patients, PMID 30165204; and Preanalytical variables that affect the outcome of cell-free DNA measurements, PMID 32393081), and discusses the isolation parameters that we found to be essential for immune cfDNA analysis.

8. The data could only partially support the aims and conclusions, and the work is in principle-of-concept stage that needs further enhancements and validations.

We agree that this is a proof of concept study, showing the potential utility of cfDNA for studying immune cell dynamics. This is further emphasized in the revised Discussion.

9. I could not find any statement on data release in the article. Please kindly add this in a revised manuscript.

In the revised manuscript we include a statement about availability of all sequencing data, and provide an example of the output of a PCR-sequencing reaction and its interpretation (detailed in Methods, and algorithm + sequence data uploaded to GitHub). In addition, we uploaded to GEO the 23 pairs of white blood cell genomic DNA + plasma cfDNA WGBS samples described in Figure 2D.